# Treatment of Undifferentiated Embryonal Sarcoma of the Liver in Children

**DOI:** 10.3390/cancers16050897

**Published:** 2024-02-23

**Authors:** Wen-Ya Lin, Kang-Hsi Wu, Chun-Yu Chen, Bei-Cyuan Guo, Yu-Jun Chang, Mao-Jen Lin, Han-Ping Wu

**Affiliations:** 1Department of Pediatrics, Taichung Veterans General Hospital, Taichung 407219, Taiwan; wylin002@gmail.com; 2Department of Pediatrics, Chung Shan Medical University Hospital, Taichung 408, Taiwan; cshy1903@gmail.com; 3School of Medicine, Chung Shan Medical University, Taichung 408, Taiwan; 4Department of Emergency Medicine, Tungs’ Taichung Metro Harbor Hospital, Taichung 43503, Taiwan; yoyo116984@gmail.com; 5Department of Nursing, Jen-Teh Junior College of Medicine, Nursing and Management, Miaoli 356006, Taiwan; 6Department of Pediatrics, National Cheng Kung University Hospital, College of Medicine, National Cheng Kung University, Tainan 70142, Taiwan; gbc628@gmail.com; 7Laboratory of Epidemiology and Biostastics, Changhua Christian Hospital, Changhua 500, Taiwan; 83686@cch.org.tw; 8Division of Cardiology, Department of Medicine, Taichung Tzu Chi Hospital, The Buddhist Tzu Chi Medical Foundation, Taichung 427213, Taiwan; 9Department of Medicine, College of Medicine, Tzu Chi University, Hualien 97004, Taiwan; 10College of Medicine, Chang Gung University, Taoyuan 333, Taiwan; 11Department of Pediatrics, Chiayi Chang Gung Memorial Hospital, Chiayi 613, Taiwan

**Keywords:** undifferentiated embryonal sarcoma, hepatic tumors, children

## Abstract

**Simple Summary:**

Undifferentiated embryonal sarcoma of the liver is a rare pediatric tumor with a traditionally poor prognosis. Dual treatment with a combination of surgery and chemotherapy has produced much improved outcomes. The sharing and consolidation of various case series and clinical experiences allow for the prompt design of appropriate treatment advice. Even in the event of unresectable or recurrent diseases, other treatment modalities such as orthotopic liver transplantation and radiotherapy can be offered. This review of treatment recommendations for rare undifferentiated embryonal sarcoma of the liver has important clinical use.

**Abstract:**

Undifferentiated embryonal sarcoma of the liver is a rare mesenchymal tumor with a highly malignant potential. It occurs almost exclusively in the pediatric population and typically has a poor outcome. Although previous studies have reported dismal prognoses, recent advances in combined treatment modalities, e.g., surgery and chemotherapy, have given cause for optimism. Even in those diseases not amenable to complete surgical resection or refractory diseases, other treatment modalities, such as liver transplant, have yielded promising results. This paper provides a review of the current treatment modalities for hepatic undifferentiated embryonal sarcoma in children.

## 1. Introduction

Hepatic tumors in pediatric patients can be classified into three major categories: epithelial tumors, mesenchymal tumors, and other malignancies [1]. Up to two thirds of pediatric hepatic tumors are malignant. Hepatoblastoma (HBL) and hepatocellular carcinoma (HCC) are epithelial tumors and are the two most common primary malignant carcinomas of the liver [2]. UESL is rare with an estimated incidence of one case per million people per year [3]. It is the third most common malignant hepatic tumor in children, accounting for 9–15% of all pediatric hepatic tumors [4]. UESL occurs mostly in children aged 6 to 10 years of age [5]. No gender predilection has been reported [3,6], although some pediatric studies have revealed a slight male predominance [7,8]. This rare hepatic malignancy typically has a high mortality. The utilization of multimodal treatment, such as combined surgical resection and chemotherapy, has dramatically improved the survival of children with UESL.

An article search of PubMed was conducted, which included the following keywords: hepatic undifferentiated embryonal sarcoma, hepatic tumor, liver transplantation, and children. Various articles were accessed and analyzed. This manuscript attempts to provide a narrative review of the various treatment regimens for pediatric UESL.

## 2. Discussion

### 2.1. Clinical Presentation and Serum Laboratory Tests

An abdominal mass with or without pain is the typical presenting symptom of pediatric UESL. This may be accompanied by systemic symptoms of fever, weight loss, or vomiting [9]. UESL typically presents as a well-circumscribed solitary mass, localized mostly over the right hepatic lobe [10]. The tumor mass has been shown to be heterogenous with mixed cystic and solid components [11]. Rapid growth with an increased size greater than 10 cm has been reported [10,12]. Rapid growth may result in wall dehiscence and subsequent tumor rupture [13]. In an analysis of 308 adult and pediatric patients with UESL, tumor rupture was an uncommon complication, accounting for only 15.7% of patients. However, tumor rupture was a significantly higher complication in children (20.9% vs. 7.1%) [12]. Extrahepatic metastasis was uncommon (20.8% of patients), with a significantly lower occurrence in children (16.8% vs. 42.1%) [12]. Metastatic diseases to the lung, pleura, peritoneum, diaphragm, and heart have been reported [12,14]. A rare case of subgaleal and epidural metastases in UESL has also been reported [15]. No distinctive laboratory findings are implicated in children with UESL. Liver function tests and inflammatory markers may be normal or mildly elevated. Tumor markers such as alpha-fetoprotein (AFP) and carcino-embryonic antigen (CEA) are typically normal [4].

### 2.2. Screening and Diagnostic Imaging

The diagnosis and management of UESL are multi-disciplinary and make use of numerous imaging tools, including ultrasonography, computed tomography (CT), magnetic resonance imaging (MRI), and 18-fluorodeoxyglucose positron emission tomography/computed tomography (^18^F-FDG PET/CT).

In the initial encounter of an abdominal mass, ultrasound is often the first-line screening test due to its relatively low cost, non-invasiveness, easy accessibility, and less demand on patient cooperation. This is especially advantageous in children as this procedure is often quicker without the need for sedation or use of anesthesia [16]. UESL exhibits a special feature of “paradoxical appearance” that may aid in early detection; this is due to its predominantly solid appearance on ultrasonography and cystic appearance on imaging with CT or MRI [13]. Ultrasonography may reveal features suggestive of malignant changes, i.e., mainly a complex mass with some solid isoechoic components and thick and irregular septations [14].

For further clarification of pediatric hepatic masses, MRI is the preferred modality. This is due to its superior soft-tissue contrast enhancement, the availability of hepatobiliary contrast agents, and the lack of ionizing radiation exposure in the acquisition of multiphase imaging. MRI is recommended in the initial evaluation of known or suspected focal liver lesions, pre-operative evaluation, and for follow-up imaging after chemotherapy [17]. Pre-operative MRI imaging allows for the detection of vascular invasion, biliary obstruction, and hilar adenopathy [4]. A large, well-defined, multiseptated mass lesion inhyperintensive T2-weighted images and isotense to hypotense appearance inT1-weighted images may be demonstrated [13].

In children for whom MRI is contraindicated, CT may be performed. CT provides the advantages of better spatial resolution, with improved delineation of small structures, and more rapid imaging time, with reduced sedation requirements [17]. CT findings revealed a cystic–solid mixed hypodense mass [18]. The lesion is usually single and well-defined, and presence of serpiginous vessels within the tumor may be uniquely found in UESL [4].

There has been increasing use of ^18^F-FDG PET/CT in the clinical management of sarcoma. As a metabolic imaging tool, it can obtain superior diagnostic information compared with use of another imaging modality alone [19]. Bone and soft tissue sarcomas have been evaluated using ^18^F-FDG PET/CT in pediatric patients. It can be used for whole body staging of sarcoma patients, including bones and metastatic lesions. It exhibited 100% accuracy, whereas technetium methylene diphosphonate (^99^Tc-MDP) studies exhibited 82% accuracy [20]. Thus, ^18^F-FDG PET/CT should be utilized in the standard diagnostic algorithm for children diagnosed with sarcomas, rendering the use of ^99^Tc-MDP bone imaging to be less mandatory [20]. Specific to UESL, two patients’ response to therapy was demonstrated by decreased PET/CT SUV, correlating with the response to therapy by pathological necrosis. The complete resolution of metabolic activity in the tumor after treatment was then demonstrated. This proved the usefulness of ^18^F-FDG PET/CT in monitoring disease progression and treatment response in pediatric UESL [21]. However, when applying these imaging modalities, great care must be taken to reduce the radiation dosage and decrease the overall cumulative radiation exposure [21].

### 2.3. Histological Diagnosis

The clinical presentation and radiological characteristics of UESL are non-specific and may share similar features with other hepatic tumors [22]. The final definitive diagnosis is largely dependent on the tissue histopathology and immunohistochemistry of the post-operative specimen. The typical macroscopic appearance of the removed surgical specimen is a larger than 10 cm solitary mass that often arises from the right hepatic lobe [4]. This mass demonstrates clear tumor borders and a fibrous pseudo-capsule with adjacent parenchymal compression. A yellow to tan cut surface is found and heterogeneous solid and cystic foci are found, and such tumors are commonly associated with areas of necrosis and hemorrhage [5]. Microscopic evaluation reveals a solid/irregular tumor architecture. The important microscopic features are hypercellular sheets of spindle cells that are highly pleomorphic with hyperchromatic nuclei and indistinct cytoplasm. Intracytoplasmic eosinophilic nuclei are also found in many cells. These medium to large spindle cells may have ill-defined borders within a myxoid matrix or fibrous stroma [13]. A high mitotic index, necrosis, frequent atypical mitoses, and apoptotic bodies demonstrating rapid cellular turnover are also characteristic features [5,13,23]. Hepatocytes and bile duct cells may be visible at the peripheral area of the tumor [18]. An immunohistochemistry study for UESL may demonstrate positivity for vimentin, desmin, CD68, B-cell lymphoma 2, and alpha-1-antitrypsin, but may be negative for hepatocyte paraffin 1, myogenin, CD34, C-kit (CD117), surfactant, anaplastic lymphoma kinase, and S100 [12]. There is no isolated specific marker for the diagnosis of UESL, but these may aid in the diagnosis by differentiating between other hepatic tumor types.

### 2.4. Treatment

Stocker et al. first reported 31 patients with UESL in 1978. This rare tumor was predominantly found in the pediatric population. The prognosis was dismal, with a median survival of less than 1 year after diagnosis [24]. Historically, treatment was based mainly on surgical resection and radiotherapy with heavy reliance on the completeness of the surgical resection for cure. Since then, multiple studies have demonstrated UESL to be a chemo-sensitive tumor with a good response to chemotherapy. Due to the rarity of pediatric UESL, most studies were based on isolated case reports or clinical experience of case series with small numbers of patients in single or multiple centers. Surgery with combined neoadjuvant or adjuvant chemotherapy was consistently reported to have a better outcome.

#### 2.4.1. Dual Treatment with Surgery and Chemotherapy

Surgery, particularly complete resection, is crucial in determining survival. Despite UESL being a chemo-sensitive tumor, chemotherapy alone is not curative. For tumors that are evaluated to be resectable at diagnosis, the best curative treatment is complete radical surgical resection with chemotherapy to optimize disease remission [25]. Lobectomy or extended lobectomy of the right liver (tumors are mostly located in the right hepatic lobe) is required for complete surgical resection without microscopic residual disease [7].

A variety of chemotherapy regimens have been explored (Table 1) [7,8,9,21,25,26,27,28,29]. Chemotherapy in the primary treatment of UESL may be divided into neoadjuvant chemotherapy and adjuvant chemotherapy. When complete resection at the initial diagnosis is not feasible, diagnostic biopsy is recommended. If this situation is due to a large tumor size, neoadjuvant chemotherapy maybe offered to shrink the tumor size to facilitate successful delayed surgical resection. In addition, the close proximity of a tumor to major hepatic vessels may pose a challenge for an initial complete surgical resection. In these cases, neoadjuvant chemotherapy may also be an option [7,8,25]. In six children with unresectable UESL with a tumor involving a large part of both lobes of the liver or invasion of the main hepatic vessels or inferior vena cava, neoadjuvant chemotherapy via two different routes were administered with success. These included pre-operative transcatheter arterial chemoembolization and systemic chemotherapy. An effective response with a reduced tumor volume, clearance of the tumor margin, and massive tumor necrosis were reported. This proposed alternative route of chemotherapy administration may serve as a promising alternative treatment in children with UESL [30].

In an earlier study by Bisogno et al., disease staging was based on the completeness of the surgical resection of the primary disease and presence of metastases. Four stages of disease staging were defined. This ranged from stage I disease with complete surgical resection, stages II and III with incomplete surgical resection and residual microscopic and macroscopic residual disease, and stage IV with the presence of metastases [9]. The TNM (Tumor, Node, and Metastasis) system has also been employed in defining clinical disease staging. T1 or T2 is determined by the invasion of contiguous organs and N0/N1 is based on an imaging evaluation of lymph nodes [31]. A treatment according to the clinical staging was then prescribed. The tumor response to treatment was evaluated after 9–12 weeks of chemotherapy. This was based on an imaging (MRI, CT) evaluation of the degree of tumor volume reduction [9]. Within the first 3 years of treatment initiation, frequent monitoring every 3 to 6 months of the treatment response with CT or MRI imaging of the primary site should be considered. Tapering to annual evaluations can be considered after 3 years of stable treatment. In addition, chest X-ray or CT chest examination are also recommended to detect pulmonary metastases [22]. The response to treatment is defined as complete response, partial response, minor response, stable disease, or progressive disease (Table 2). A complete response is characterized by the complete disappearance of the disease. A partial response is defined as a tumor volume reduction of more than two-thirds and a minor response is defined as a tumor volume reduction of one-third to two-thirds of the initial size. A tumor volume reduction of less than one-third is considered to be a stable disease. A progressive disease is defined as an increased tumor size or development of new lesions. The response may sometimes be difficult to determine using imaging studies as the cystic component of the tumor may not show a reduction in size, but the histological examination of a surgically removed mass may reveal a high rate of cell necrosis. Thus, the response evaluation included clinical and histological findings [31,32]. More recent studies have also suggested the future role of ^18^F-FDG PET/CT in assessing initial disease staging, response to treatment, and detection of recurrence [19,20,21].

The dramatic improvement in long-term survival (70%) with combined surgical and chemotherapy treatment has sparked further interest. A rhabdomyosarcoma-based chemotherapy regimen was employed in the study by Bisogno et al. Tumor shrinkage following chemotherapy was found in nearly two-thirds (62%) of children with UESL [9]. This favorable survival outcome was echoed by another study of 25 children also receiving a similar rhabdomyosarcoma-based chemotherapy (72% survival) [26]. These studies demonstrated the chemo-sensitive nature of UESL and paved the way for future treatment designs.

Kim et al. grouped children with UESL according to the surgical resectability of the primary tumor. Those with a lesion involving both lobes of the liver including the caudate lobe, a sufficiently large tumor size precluding a safe surgical resection, and distant metastasis were deemed initially unresectable. These children then underwent diagnostic biopsies or partial resection with residual tumor, with complete resection after neoadjuvant chemotherapy. Chemotherapy according to the third Intergroup Rhabdomyosarcoma Study (IRS III) was given, with regimen 35 for those with initially unresectable diseases and regimen 38 for resectable diseases. Four children (out of a total of six) successfully completed adjuvant chemotherapy, which was initiated 2 weeks after surgery. A total of five children remain well without tumor recurrence at 40, 45, 48, 60, and 122 months, respectively [8].

An analysis of six children with UESL by Techavichit et al. revealed an overall survival rate of 67%. All children with a favorable outcome had complete tumor removal with free surgical margins after partial hepatectomy and orthotopic liver transplantation (OLT) [27].

In the study by Upadhyaya et al., children with UESL received chemotherapy mostly according to the malignant mesenchymal tumor study, although for a few patients, the international Childhood Liver Tumour Strategy Group (SIOPEL) 3 treatment regimen was used. Interestingly, among this group of patients with pediatric hepatic sarcoma, children with UESL had better overall survival (72%) in comparison to those with other hepatic sarcoma types, including rhabdomyosarcoma, rhabdoid tumor, and interdigitating dendritic cell sarcoma. One of the mortalities in this series was a child initially misdiagnosed with hepatoblastoma, which was later revised to a diagnosis of UESL. The patient receiving SIOPEL chemotherapy showed a poor response and recurrent disease with lung metastasis [28].

Survival was poorer at 57% in a study by Webber et al., and the chemotherapy agents used were based on the sarcoma malignancy protocol and those of the hepatic malignancy protocol. The earlier patients received chemotherapy based on the hepatic malignancy protocol, including cisplatin, doxorubicin, 5-fluorouracil, and vincristine. The drugs used in the sarcoma malignancy protocol included etoposide, ifosfamide, vincristine, doxorubicin, and cyclophosphamide. The mortalities were mostly due to tumor recurrence [25].

In a study by Ismail et al., a further improvement in survival up to 90% was reported with combined chemotherapy and surgery. The single mortality reported in this study was a patient initially misdiagnosed with hepatoblastoma and treated with inappropriate chemotherapy; a poor response and disease progression to mortality was observed [7].

The best survival results, up to 100% survival (*n* = 5), were reported by Plant et al. Four children received complete surgical resection and a ifosfamide-based and doxorubicin-based regimen. One child received OLT for recurrent disease. This multi-modal approach included individualized treatment modification, such as radiotherapy for metastases and timely OLT referral and evaluation [21]. With a similar treatment regimen of 5 to 6 cycles of a predominantly ifosfamide and doxorubicin-based regimen, Walther et al. also demonstrated an overall survival of 100% [29]. The two aforementioned studies demonstrating 100% survival were also the more recently reported studies, demonstrating a gradual improvement in outcomes as the clinical research and knowledge advance. Both studies had cases of patients receiving timely successful OLT after recurrent or refractory disease with a good outcome. In the treatment of pediatric UESL, metastasis at the initial diagnosis did not always indicate a poor prognosis. A chemotherapy response at the metastatic site has been reported despite a poor response at the primary site [28].

Overall, the chemotherapy agents employed for the treatment of UESL are mostly based on those used in the rhabdomyosarcoma or soft tissue sarcoma protocol, rather than those applied in the hepatic malignancy protocol. Most studies have found a better response to sarcoma-based chemotherapy regimens, with poorer outcomes in those treated with hepatoblastoma-based chemotherapy [7,28]. Multiple drugs with activity against UESL include a combination of ifosfamide, etoposide, carboplatin, doxorubicin, flavopiridol, irinotecan, vincristine, cisplatin, cyclophosphamide, dactinomycin, and 5-fluorouracil [12].

#### 2.4.2. Liver Transplantation

Starzl et al. performed the first liver transplantation in 1963. Since then, liver transplantation has been increasingly recognized as the treatment of choice for a variety of diseases. These include cirrhosis, decompensated disease, acute liver failure, and certain hepatic malignancies [33]. In the modern era, the evolution of liver transplantation techniques has allowed their clinical application in a variety of pediatric malignant and non-malignant liver diseases.

Orthotopic liver transplantation involves the removal of a diseased native liver, and substituting it with a normal liver (or part of one) taken from a deceased or living donor. It may serve as a therapeutic option in children with UESL with unresectable hepatic lesions due to anatomical reasons (proximity to vital structures, invasion of major vessels, multifocality, large size) or refractory to neoadjuvant chemotherapy (no shrinkage in tumor size) and recurrent tumors despite initial effective primary treatment [34,35]. Previously, liver transplantation for malignancy was discouraged due to concerns of primary malignancy recurrence and morbidity/mortality with long-term immunosuppression. This has changed with numerous reported cases of successful OLT. An increasing number of studies have reported evidence supporting the notion of “transplant oncology”, which entailed the integration of transplant surgery and surgical oncology. An aggressive multidisciplinary approach to oncology has provided additional treatment options involving OLT in the treatment of hepatobiliary malignancies, including UESL [36].

The studies employing primary and salvage OLT in UESL treatments is listed in Table 3. Plant et al. reported a successful OLT in a patient with refractory UESL, with an overall survival duration of 61 months from diagnosis [21]. Walther et al. demonstrated similar encouraging results of OLT in three children with unresectable or recurrent disease [29]. A larger systematic review of 28 adult and pediatric patients evaluated the practicality of OLT as an alternative treatment for UESL. In patients receiving primary OLT (82%) and salvage OLT (18%), the survival rate was up to 96%, with a median follow-up of 28.5 months [35]. Therefore OLT may be a viable treatment option for locally unresectable or recurrent UESL.

Comparisons of liver transplantation outcomes in children with UESL, children with HBL, and non-malignant disease revealed no significant differences in the 1-year and 5-year survival rates. The recipient and graft survival rates were also not significantly different for children with UESL (compared with HBL and non-malignancy cohorts). The median follow-up after transplant was 7.0 (2.9–11.2) years for UESL children, 4.8 (1.1–10.0) years for HBL, and 7.1 (1.9–14.4) years for non-malignancy indications (*p* < 0.001). No death due to UESL recurrence was found [34].

Schluckebier et al. also reported a successful primary OLT in a 10-year-old child with UESL and complete remission was found at 3 years after transplantation. The study noted the use of tacrolimus-based immunosuppression after OLT, and a combination of calcineurin inhibitors and mTOR inhibitors was initiated at around 1 to 3 months post-transplantation (after adequate wound healing). The antiproliferative characteristics of mTOR-inhibitors have been shown to improve survival following transplantation in adult hepatocellular carcinoma patients. This study revealed a similar successful result in a child with UESL who received primary OLT [37].

Thus, OLT may be considered an effective and safe primary treatment for children with unresectable and chemotherapy-refractory UESL. Also, in those with recurrent diseases, OLT may serve as salvage treatment. The early detection of recurrent disease is important and a timely referral to a liver transplantation team is essential.

#### 2.4.3. Radiotherapy

Over the years, the role of radiotherapy has diminished in the treatment of UESL. Radiotherapy is used in about 15% of cases [12]. The decision to employ radiotherapy has been mainly based on the individual requirements of the patient. It has been used to treat metastases in the lung and paraspinal area [21]. Treatment with radiotherapy for disease relapse has also been offered [38].

### 2.5. Genetic Abnormalities and Associated Disease

New genetic studies have revealed associations between genes and human sarcomas. These included Budding Uninhibited By Benzimidazoles (BUB) which are group of genes encoding for proteins that play a pivotal role during mitosis. Dysregulated BUB expression and mitosis is associated with variety of malignancies, including sarcoma. Higher expression levels of BUB1, BUB1B, and BUB3 were found in sarcoma samples and were associated with lower overall and disease-free survival in sarcoma patients. BUB1, BUB1B, and BUB3 may serve as a future potential treatment targets and prognostic biomarkers for patients with sarcomas [39]. In addition, the mRNA expression levels of the four GINS family members (GINS1, GINS2, GINS3, GINS4) were all higher in sarcoma tissue. An increased expression of GINS genes was associated with poor overall survival, disease-free survival, and relapse-free survival and could serve as prognostic biomarkers [40]. However, similar genetic studies specific to UESL are not yet available and require further research.

UESL and mesenchymal hamartoma (HMH) are two separate clinical entities with overlapping clinicopathological characteristics and similar chromosomal aberrations (19q13.4) [5,14]. HMH is a benign tumor, with the majority of children presenting before the age of 2 years, and a slight male predominance has been observed [4,41]. UESL has been reported to arise from HMH, with possible malignant transformation years after incomplete HMH resection [42,43]. The mechanism of tumorigenesis is still unknown. The loci involving the MALAT1 gene on chromosome 11 and a gene-poor region termed MHLB1 on chromosome 19 have been shown to be involved in translocation in cases of UESL arising from HMH. In an attempt to explore the genetic events of HMH tumorigenesis, capture-based next-generation sequencing (NGS) targeting these loci was performed. The identified chromosome rearrangements included translocation t(11,19)(q13.1;q13.42) involving the MALAT1 gene; the translocation t(2,19)(q31.1;q13.42) involving AK023515, an uncharacterized noncoding gene; and the inversion inv(19,19)(q13.42;q13.43) involving the PEG3 gene encoding a Kruppel-type zinc-finger protein [44]. Eleven children with UESL were identified and five cases were found to be associated with HMH. The immunohistochemistry studies all revealed positivity for vimentin, diastase-resistant periodic acid–Schiff stain, and alpha-1 antitrypsin. A chromosomal analysis of three UESL patients, two of which had HMH, revealed no involvement of 19q13.4. Thus, much work is still required to determine the presence of other genetic aberrations in this UESL/HMH association [45].

### 2.6. Outcome

UESL are malignant tumors with a poor survival rate of less than 37.5% [24,46]. The implementation of a multimodal treatment approach dramatically improved survival outcomes. Shi et al. utilized a multi-institutional database (1998–2012) and evaluated the disease outcomes of 103 children with UESL. The 5-year overall survival was 86%, with significantly improved outcomes in those given a combined treatment (92%). A significantly better survival was also found in children who received sectionectomy or hemihepatectomy. Complete surgical resection remains the indisputable cornerstone of UESL treatment. In five selected recipients of isolated surgical resection with negative margins (one sectionectomy, two hemihepatectomies, one trisectionectomy, and one OLT), the overall survival was 100%. Surprisingly, a positive tumor margin after surgical resection was not associated with poorer outcomes in this study. Residual UESL can thus be effectively eliminated with combined chemotherapy [3].

Wu et al. identified 308 patients (71.1% children) from multiple case reports and case series. The 5-year survival rates (overall survival 79.9% in children vs. 49.5% in adults) and disease-free survival (87.0% in children vs. 51.1% in adults) of the children were significantly higher than those of the adults. The combined partial hepatectomy and chemotherapy was found to be significantly associated with improved overall and disease-free survival. A significantly higher number of children received a combination of surgery and chemotherapy (67.0%ofchildren vs. 33.3%ofadults), which could also possibly explain the better survival in children [12]. A retrospective study by Ziogas et al. reported similar findings, with more children receiving chemotherapy than adults (92.7% vs. 65.9%; *p* < 0.001) and a significantly better 5-year overall survival rate being found in the children (84.4% vs. 48.2%) [47]. The initial tumor size and extrahepatic metastatic disease were not significantly associated with decreased survival. Thus, the presence of a large tumor size or metastases does not imply an inevitable aggressive course with a poor prognosis, as patients can still benefit from appropriate treatment advice [12,31].

Guérin et al. also evaluated the outcomes of 65 UESL patients up to 21 years of age treated according to the European soft tissue sarcoma protocol, and the 5-year overall survival and event-free survival rates were 90.1% and 89.1%, respectively. The chemotherapy regimens used were mainly alkylating agent (vincristine, cyclo, or ifosfamide and actinomycin)-based regimens. Up to two thirds of patients receiving chemotherapy had reduced tumor sizes. The use of neoadjuvant chemotherapy increased the negative margins after delayed surgical resection and decreased tumor spillage. The beneficial role of anthracycline-based agents was not as well defined in this study. Anthracyclines were employed only in a few patients with more advanced diseases, with no difference in overall survival or event-free survival [31].

## 3. Conclusions

The past treatment of UESL with surgery alone or surgery in combination with radiotherapy has led to poor outcomes. Currently, the dual use of surgery and chemotherapy is recommended, with much more optimistic prognoses. A multi-modal approach is suggested with tailor-made modifications according to individual needs.

Surgery and, where possible, complete radical surgery, remain the indisputable core of treatment advice. This has remained unchanged. An emphasis on improved primary surgical resection techniques, the use of neoadjuvant chemotherapy to establish early tumor shrinkage, optimizing subsequent surgical resection, and timely OLT treatment advice have all contributed to improved prognoses. Increasing evidence has revealed liver transplantation to be an effective and safe alternative therapeutic option for unresectable, refractory, and recurrent UESL disease. In fact, compared with liver transplantation for other pediatric HBL and non-malignant diseases, survival is similar and no recurrent disease has been found.

In addition, multiple studies have demonstrated UESL to be a chemo-sensitive tumor with a good response to appropriate chemotherapy regimens. Thus, chemotherapy has been proven to be effective. The presence of a large tumor size and metastases does not necessarily predict a poor prognosis, and should not deter subsequent treatment recommendations. For those in advanced stages of the disease, palliative care including analgesia, palliative chemotherapy, or radiotherapy aiming to improve quality of life are also crucial. Specific guidelines for palliative treatment for UESL patients were not found. However, palliative treatment should be individualized to meet each patient’s physical, psychological, and spiritual needs.

Due to the rarity of cases, there is currently no established chemotherapy protocol for UESL. However, protocols based on chemotherapy for pediatric rhabdomyosarcoma or soft tissue sarcoma has led to promising results. Both regimens with a predominantly alkylating agent or an anthracycline agent have been demonstrated to be effective, and therefore the choice of the specific agent may be influenced by disease severity and chemotherapy side effects.

All in all, the findings reported above indicate that the outcomes of children with UESL are no longer dismal. With improved survival, especially for children with UESL, more focus should be placed on guidance for long-term surveillance and methods of minimizing treatment-related adverse side effects. Vigilant monitoring of the treatment response should be performed to allow for the early detection of refractory cases and the timely introduction of OLT to further optimize the outcomes of children with UESL.

## Figures and Tables

**Table 1 cancers-16-00897-t001:** Chemotherapy employed in the treatment of UESL.

Study	Year, Case Number	Neoadjuvant Chemotherapy	Adjuvant Chemotherapy	Chemotherapy Regime and Combination	Survival
Bisogno et al. [9]	1979–1995 10 children	Stage III: CAV for 4 courses	Stage I, II, III: VAC + CAV alternating cycle for 12 courses	Italian Rhabdomyosarcoma Group (RMS) 79 VAC, CAV	70.6%
		Stage III: VAIA	Stage I, II, III: IVA for 9 courses	Italian Rhabdomyosarcoma Group (RMS) 88 IVA, VAIA	
	1981–1998 7 children	VACA or VAIA for 2–3 course according to clinical stage		German Cooperative Weichteilsarkom Studie (CWS) 81 VACA German Cooperative Weichteilsarkom Studie (CWS) 86, and 91 VAIA	
Ismail et al. [7]	1981–2012 10 children	7 children (6 with response with radiological imaging and pathological exam): 4 with CAV/ETIF/IF + ADM 2 with CYVADIC 1 with CDDP/PLADO	10 children	CYVADIC, CAV, VAC CAV/ETIF/IF + ADM, IVADO, VP-16/CDDP, Vinorelbina/CTX IVADO, VP-16/CDDP, Vinorelbina/CTX CDDP/PLADO PLADO, CAV	90% (50–222 months follow-up)
Webber et al. [25]	1987–1998 7 children	3 children (2 with significant tumor shrinkage)	7 children	-Based on sarcoma malignancy protocol or hepatic malignancy protocol AEIV, ACEIV, ACEIP, AFPV, FPV, AFP, ACFV	57% (4 children with remission at 19, 27, 68, and 150 months)
Murawski et al. [26]	2007 (end of study period) 25 children	13 children	20 children	CWS-96: Low-risk group: vincristine, dactinomycin Standard-risk group: ifosfamide, vincristine, dactinomycin High-risk group: vincristine, dactinomycin, ifosfamide, doxorubicin or carboplatin, epirubicin, vincristine, dactinomycin, ifosfamide, etoposide (randomized) CWS-2002P: Standard-risk group: ifosfamide, vincristine, dactinomycin High-risk group: ifosfamide, vincristine, dactinomycin, doxorubicin CYVADIC	72%
Upadhyaya et al. [28]	1988–2007 11 (19) children with UES	10 children	-	Malignant mesenchymal tumor (MMT) 89, 95, and 98 International Childhood Liver Tumour Strategy Group (SIOPEL) 3—initially misdiagnosed	72%
Kim et al. [8]	1990–2000 6 children	Initial unresectable disease-regimen 35 for 1 year	Resectable disease-regimen 38 for 1 year	Third Intergroup Rhabdomyosarcoma Study (IRS III) Regimen 35: pulsed VAC + radiotherapy beginning on day 42 + CDDP cisplatin with mannitol + adriamycin (days 14, 21, 35, 42, 49, and 56) Regimen 38: VADRC-VAC + CDDP	83% (5 children with remission at 40, 45, 48, 60, and 122 months)
Techavichit et al. [27]	1993–2014 6 children	5	6	ICE (6 cycles) IE Ifos/Dox (5–6 cycles) VAC and IE	67%
Plant et al. [21]	2001–2011 5 children	0	5 children	-Ifosfamide, doxorubicin for 5 cycles (3 children) -Italian Rhabdomyosarcoma Group (RMS) 88 (1 child) -Cisplatin, doxorubicin for 5 cycles (1 child)	100% (21–68 months)
Walther et al. [29]	2002–2012 6 children	4 children -Ifosfamide, doxorubicin → shift to vincristine, doxorubicin, cyclophosphamide for 3 cycles (1 child) -Ifosfamide, doxorubicin for 5 and 6 cycles (2 children) -Vincristine, actinomycin D, ifosfamide. doxorubicin for 9 cycles, and recurrent disease with etoposide, carboplatin for 2 cycles and VAIAfor 9 cycles (1 child)	6 children -Ifosfamide, doxorubicin for 5 cycles, ifosfamide for 1 cycle (1 child) -Vincristine, doxorubicin, cyclophosphamide alternating with ifosfamide, etoposide for 17 total cycles (1 child) -Vincristine, doxorubicin, cyclophosphamide for 3 cycles (1 child) -Vincristine, actinomycin D, cyclophosphamide for 4 cycles (1 child) -Ifosfamide for 1 cycle (1 child) -VAIA for 6 cycles (1 child)		100% (meanfollow-up of 35 months)

ACEIV: adriamycin, cyclophosphamide, etoposide, ifosfamide, vincristine. ACEIP: adriamycin, cyclophosphamide, etoposide, ifosfamide, cisplatinum. ACFV: adriamycin, cyclophosphamide, 5-fluorouracil, vincristine. AEIV: doxorubicin, etoposide, ifosfamide, vincristine. AFP: adriamycin, 5-fluorouracil, cisplatinum. AFPV: adriamycin, 5-fluorouracil, cisplatinum, vincristine. CAV: cyclophosphamide, doxorubicin, vincristine. CYVADIC: dacarbazine, vincristine, doxorubicine, cyclophosphamide. FPV: 5-fluorouracil, cisplatinum, vincristine. ICE: ifosfamide, carboplatin, etoposide. IE: ifosfamide, etoposide. Ifos/Dox: ifosfamide, doxorubicin. IVA: ifosfamide, vincristine, actinomycin D. VAC: vincristine, actinomycin D, cyclophosphamide. VACA: vincristine, actinomycin D, cyclophosphamide, Adriamycin. VAIA: vincristine, actinomycin D, ifosfamide, doxorubicin.

**Table 2 cancers-16-00897-t002:** Treatment response in UESL.

Complete response	Complete disappearance of disease (confirmed clinically/histologically)
Partial response	>66% tumor volume reduction
Minor response	33–66% tumor volume reduction
Stable disease/no response	<33% tumor volume reduction
Progressive disease	Increase in tumor size or appearance of new lesions

**Table 3 cancers-16-00897-t003:** OLT in the treatment of UESL.

Study	Study Type, Year, Number of Cases	OLT Indication	Survival
Plant et al. [21]	Case series, 2001–2011, 1 child	Salvage OLT: received chemotherapy and surgery with tumor recurrence at liver	Disease-free for 37 months, overall survival duration of 61 months
Walther et al. [29]	Case series, 2002–2012, 3 children	2 children with neoadjuvant chemotherapy and primary OLT, 1 child with liver tumor recurrence and salvage OLT	Survival of 100% at mean follow-up of 35 months
Schluckebier et al. [37]	Case report, 1 child	Primary OLT	Survival and remission 36 months after transplantation
Babu et al. [35]	Systematic review, inception of database to end of December 2018, 28 adults and children	82% primary OLT 18% salvage OLT	Survival of 96%, median follow-up of 28.5 months

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
