# Peer review of "Treatment of Undifferentiated Embryonal Sarcoma of the Liver in Children"

_cancers, 2024, doi:10.3390/cancers16050897_

Round 1
Reviewer 1 Report
Comments and Suggestions for Authors
This is a review article of reports on undifferentiated embryonal sarcoma of the liver, from diagnosis to treatment. While reports are limited due to the rarity of this tumor, it is commendable that the data was collected and compiled extensively.
#1. One point is that while the article describes in detail the treatment options for curative purposes, there is little information on treatment options in the palliative setting. Bearing in mind the poor prognosis of this tumor, the importance of palliative treatment cannot be disputed. I would appreciate additional information on treatment options in the advanced stages of the disease.
#2. Recently, with the widespread use of NGS, more and more is being learned about genetic abnormalities. I would like to request additional information on reports of genetic abnormalities in this tumor after an additional survey.
Comments on the Quality of English LanguageThere are no significant problems with the English used in this paper.
Reviewer 2 Report
Comments and Suggestions for Authors
The manuscript provides a comprehensive overview of UESL, including its clinical presentation, diagnostic criteria, and treatment modalities. This review has been well-written. The inclusion of recent studies and systematic reviews strengthens its relevance and utility for practitioners in the field.
Reviewer 3 Report
Comments and Suggestions for Authors
Dear Authors
The topic of the article is very interesting.
I have some comments and questions.
1. Does UESL metastasize to bones and to bone marrow?
2. Laboratory test - does LDH increase in UESL?
3. What is the role of USG with contrast in diagnosis of UESL?
4. What is OLT?
5. What are the prognostic factors in UESL?
6. lines 361-363 - significantly higher number of children received a combination of surgery and chemotherapy 67% vs 33,3% - is it comparison to adults?
7. what is the role of radiotherapy in treatment of UESL in children and adults?
Reviewer 4 Report
Comments and Suggestions for Authors
1) Table 1 is very full and caption is mandatory: it is advisable to use abbreviations in table (i.e. for chemotherapeutic regimens for which the extended regimes can be written in caption).
2) In paragraph 2.2 “Diagnostic Imaging” I encourage authors to add, if availables, some emblematic figures with captions showing the performance of ultrasonography, CT, MRI, and 18F-FDG PET/CT in detecting UESL.
3) In paragraph 2.4.1. “Dual Treatment with Surgery and Chemotherapy” I suggest that you do not use table 2 about disease staging of UESL and table 3 about treatment response in UESL. These topics are interesting but I think you should deepen them in introduction, for example.
4) Paragraph 2.4.1. “Dual Treatment with Surgery and Chemotherapy” gets repetitive. A better built table 1 is the best thing. In the main text, the studies are usually presented in such a detail that is not actually necessary. It is not attracting to report all the numbers and results that a study has achieved. Only the most important and straightforward part of a study in needed to be reported.
5) I encourage authors to add a well built table with general characteristics, numbers and results of included studies in paragraph 2.4.2. “Liver Transplantation” and in paragraph 2.6 “to report in the main text the most important part of the studies.
Round 2
Reviewer 1 Report
Comments and Suggestions for Authors
The authors have responded appropriately to the reviewers' comments and have made revisions to the manuscript.
Reviewer 3 Report
Comments and Suggestions for Authors
I don't have.
Reviewer 4 Report
Comments and Suggestions for Authors
The authors have revised the manuscript as requested.